# Dynamic Pricing on E-commerce Platform with Deep Reinforcement Learning

## Abstract

In this paper we develop an approach based on deep reinforcement learning (DRL) to address dynamic pricing problem on E-commerce platform. We models real-world E-commerce dynamic pricing problem as Markov Decision Process. Environment state are defined with four groups of different business data. We make several main improvements on the state-of-the-art DRL-based dynamic pricing approaches: 1. We first extend the application of dynamic pricing to a continuous pricing action space. 2. We solve the unknown demand function problem by designing different reward functions. 3. The cold-start problem is addressed by introducing pre-training and evaluation using the historical sales data. Field experiments are designed and conducted on real-world E-commerce platform, pricing thousands of SKUs of products lasting for months. The experiment results shows that, on E-commerce platform, the difference of the revenue conversion rates (DRCR) is a more suitable reward function than the revenue only, which is different from the conclusion from previous researches. Meanwhile, the proposed continuous action model performs better than the discrete one.

## 1    Introduction

Dynamic pricing, often referred to revenue management as well, is to adjust prices according inventories left and demand response observed from time to time to maximize revenue. It has drawn great attentions during the past decades since the deregulation of the airline industry in the 1970s. Weatherford & Bodily (1992) and Talluri & Van Ryzin (2006) gave overviews of the research that has been done in the field of perishable-asset revenue management, which is a field that combines the areas of yield management, overbooking, and pricing.

During the recent development of business, many industries have become more active in revenue management. Riding-share platforms like Uber has implemented dynamic pricing strategy, known as 'surge' pricing and Chen & Sheldon (2016) showed that it has significant impact on motivations for more driving times. Retailers like Zara has implemented systematic dynamic markdown pricing strategy. Caro & Gallien (2012) studied the clearance pricing for Zara to increase the revenue generated by each item while maintaining a large number of items to sell. Kroger is now testing electronic price tag at one store in Kentucky (Nicas (2015)).

Online retailers have a stronger desire for dynamic pricing strategies due to more complex operations. Amazon.com sells 356 million products (562 million now). Walmart.com sells 4.2 million products according to a 2017 estimate[1]. Taobao.com, the biggest E-commerce platform in China, sells billions of products at present. Operation specialists have to set prices for these items periodically to remain competitive, which will be mission impossible when the number of items goes this high. As a result, Amazon has implemented automatic pricing systems and it is reported that Amazon.com can change prices every 15 minutes[2]. Chen et al. (2016) studied the pricing strategies for Amazon.com empirically and derived significant factors for pricing.

In this paper, we proposed a reinforcement learning approach to address the dynamic pricing problem for online retailers. The scenario we consider is how to dynamically price for different goods on Tmall.com, the largest business-to-consumer retailer in China, spun off from Taobao.com. There

---

[1]https://www.scrapehero.com/how-many-products-does-walmart-com-sell-vs-amazon-com
[2]https://www.whitehouse.gov/sites/default/files/docs/Big_Data_Report_Nonembargo_v2.pdf

are many difficulties for pricing on such an E-commerce platform. First, the market environment is impossible to be quantified. Revenues for the same product could change dramatically due to unpredictable fluctuation of the daily customer traffic, the change of other products' prices or even comments from the previous buyers. Second, it would lead to non-convergence policies if the reward function is not set properly under such complicated environment. Third, it is not applicable to apply the pricing models online directly, since a slightly inappropriate price online could quickly cause large capital loss. Therefore, the pre-training and evaluation aiming this cold start problem becomes necessary. Fourth, unlike recommendation system, it is impossible to do online A/B testing, because it is illegal to expose different prices at the same time to different customers. That is an obstacle to evaluate the performances of different pricing policies during the field experiment.

To overcome these difficulties, we propose a framework for dynamic pricing with DRL to optimize long-term revenue. This paper has several contributions. First of all, we are the first to apply DRL for both discrete and continuous pricing problem on real-world E-commerce platform, rather than traditional discrete one using Q-learning (Maestre et al. (2018)) or within simulated environment. Second, we found that the revenue conversion rate and its difference are more suitable as the reward function rather than the revenue (Maestre et al. (2018), Kim et al. (2016), Schwind & Wendt (2002)) due to its convex nature. Third, very few assumptions are made: the demand function is not defined in advance and not necessarily time-invariant; the revenue function is not necessarily convex or concave; the customer behaviour could be strategic. Finally, we carry out large-scale field experiments lasting for months with thousands of SKUs products priced by our DRL models. Our field experiments prove the effectiveness of our dynamic pricing framework on E-commerce platform, which is the first work of this kind with such achievement.

The remaining of this paper is organized as follows: The next section lists some related work in dynamic pricing problem. Section 3 introduces approaches we designed for dynamic pricing, where the problem is modeled as a Markov Decision Process model. Both discrete pricing action model and continuous pricing action model are proposed. In section 4, the results from both offline and online experiments are introduced, which validate our approach of the problem. The conclusions and future work directions are summarized in section 5.

## 2 LITERATURE REVIEW

Much research has been done in dynamic pricing for decades. We refer to den Boer (2015) for a comprehensive review for recent developments. It combines two research fields: (1) statistical learning, specifically applied to the problem to estimate demand and (2) price optimization. Most of previous research has focused on the cases where a functional relationship between price and demand is assumed to be known to decision makers. Cournot (1897) is acknowledged to be the first to mathematically describe the price-demand relation of products and solve the mathematical problem to achieve the optimal revenue. However, it assumed that the relationship is static over time which usually does not hold true in reality. Evans (1924) assumed that the demand is not only a function of price, but also the time-derivative of price, leading to a dynamic demand function of price over time. Kamrad et al. (2005) introduced a stochastic model to capture demand uncertainty while optimizing the prices. Gallego & Van Ryzin (1994) considered constraints like limited inventories and a finite planning horizon.

In practice, it is often difficult to describe the demand beforehand. Much recent research focuses on the dynamic pricing with unknown demand function. Researchers first addressed the problem by parametric approaches. Bertsimas & Perakis (2006) assumed parametric families of demand functions to be learned over time. Farias & Van Roy (2010) proposed an approach to learn from the historical purchase data. Harrison et al. (2012) utilized Bayesian dynamic pricing policies to address demand uncertainties. However, revenue may depart from the optimal due to mis-specifying the demand family. Therefore, much recent research mainly revolves around non-parametric approaches. Besbes & Zeevi (2009), Besbes & Zeevi (2015), Wang et al. (2014) looked deep inside learning while earning approaches; and Wang et al. (2014) announced to have the smallest gap so far. However, they all assumed the revenue function is strictly concave and differentiable, which could not hold true in E-commerce retail industry as shown in section 3.1.

With the development of computation, reinforcement learning (RL) is introduced to address dynamic problems. Kephart et al. (2000) demonstrated the possibility of using Q-learning to express

anticipated future-discounted profits for possible prices, forming a so called pricebot to adjust prices in response to changing market conditions. Schwind & Wendt (2002) used Temporal Difference for Information products' dynamic pricing from a yield management view. Raju et al. (2003) formulated single seller and two sellers dynamic pricing problems and employ different RL algorithms in a simulated context. Kutschinski et al. (2003) used different types of asynchronous multi-agent RL methods to determine the competitive pricing strategy in the market scenario. Vengerov (2007) and Kim et al. (2016) utilized reinforcement learning to optimize prices in the energy market. Maestre et al. (2018) suggested Q-learning with neural networks approximation to maintain revenue while improving fairness in a simulated environment. All these previous works, however, are carried out in simulations with simplified market settings, where the reward defined with revenue work out well. And DNNs are only used as approximators for discrete prices. It is not the case for real-world market.

# 3 METHODOLOGY

We now consider how to build up decision making models for the dynamic pricing problem we discussed above. We first represent the problem as a Markov Decision Process (MDP). The agent periodically changes prices of the products as its action after observing environment state. The new environment state could then be observed and the reward could also be received. Each pricing episode reaches its end if the product is out of stock. The model is pre-trained by historical sales data and previous specialists' pricing actions, which is also used for offline evaluation. The framework is shown in Figure 1.

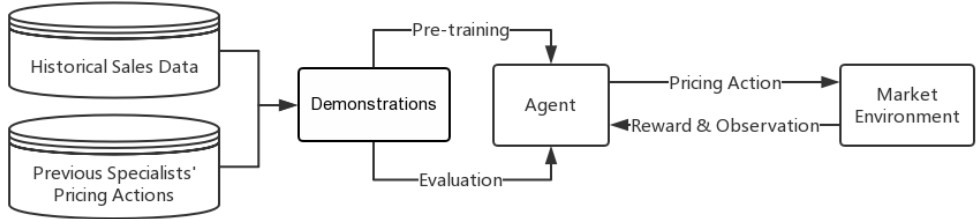

Figure 1: dynamic pricing framework using DRL with demonstrations on E-commerce platform.

## 3.1 PROBLEM FORMULATION

We mainly consider two kinds of dynamic pricing applications for E-commerce platform, named **markdown pricing** and **daily pricing** in the rest of this work. These two kinds of pricing applications could cover most dynamic pricing scenarios on E-commerce platform. For both markdown pricing and daily pricing, we define $n$ products labeled by $i = 1, 2, ..., n$ individually. The prices are decided to be modified or kept at discrete time steps $t = 1, 2, ..., T$, while between these time steps the prices will be fixed referring to the model of market in Kutschinski et al. (2003), Madhavan (2000) and O'hara (1995). The distance between two time steps is define by a hyperparameter $d$. For markdown pricing the supply is limited, so the pricing process for certain product reaches its end if it is out of stock, while the supply in daily pricing is regarded as unlimited. The dynamic pricing process on E-commerce platform is formed as Markov Decision Process (MDP) referring works from Kim et al. (2016), Vengerov (2007), Raju et al. (2003) etc. At each time step $t$, the pricing agent observes $s_{i,t}$ describing the state for product $i$, and takes an action $a_{i,t}$. Then the agent receives the reward $r_{i,t}$ for that action as well as the observation for the new state $s_{i,t+1}$. These four elements form the transition $(s_{i,t}, a_{i,t}, r_{i,t}, s_{i,t+1})$, which could be simplified as $(s, a, r, s')$.

Intuitively we want a small $d$ to make pricing actions reacting in time or archiving continuous pricing. However, precisely describing the change of the environment may need a certain time period for observing. Changing price rapidly could also break *price image* for the product and even cause credit issue on E-commerce platform. Since our experiments would change the prices online, we set this period carefully after discussions with professional pricing managers. In the rest of this work, pricing period $d$ is settled as one day. Therefore, time step $t$ also represents day $t$.

**State space.** Here in our model, each product $i$ is priced separately, with four different groups of features at time step $t$ to describe the state $s_{i,t}$: price features, sales features, customer traffic features and competitiveness features. Price features contain the actual payment for this product, the discount rate, the coupons, etc. Sales features contain the sales volume, revenue, etc. Customer traffic features contain the time the page of the product $i$ has been viewed (PV), the number of unique visitors viewed the product (UV), the number of buyers for product $i$, etc. The comments and the states of the similar products contribute to competitiveness features. The explanations for some core features are given in Table 1 (in Appendix).

**Action space.** We also define the action space for each product $i$ separately. We use the maximum price $P_{i,max}$ and the minimum price $P_{i,min}$ of the product $i$ during a certain number of periods in the history to define the upper bound and lower bound. It assumes that the pricing framework should not output a price out of this area. The pricing space could be discrete or continuous for different applications. When it is discrete, each action stands for a price range.

**Reward function.** We compared different ways to define the immediate reward $r_{i,t}$. The revenue is not suitable as we mentioned above, the customer traffic could obviously change without forewarning and then influence the revenue. Therefore, there may not be a clear and explainable relationship between price and revenue like in traditional retail industry. Defining reward function using revenue could severely mislead the agent and end up with non-convergence. But on E-commerce platform, there are links between price and revenue conversion rate, $r_{i,t} = revenue_{i,t}/uv_{i,t}$, where $uv_{i,t}$ represents the number of unique visitors viewed the product $i$ between time step $t$ and $t-1$. In some part of this work, we also use profit conversion rate, dividing by $u_{i,t}$, if we have the knowledge of the inventory cost.

To prove this idea, we analyze different categories of products about the distribution of revenues and their conversion rates on different prices. Here, we demonstrate the results for over 3300 SKUs of shampoos and over 4800 SKUs of candies selling on Tmall.com, about their average revenue and revenue conversion rate in different price levels within 3 months in Figure 2.

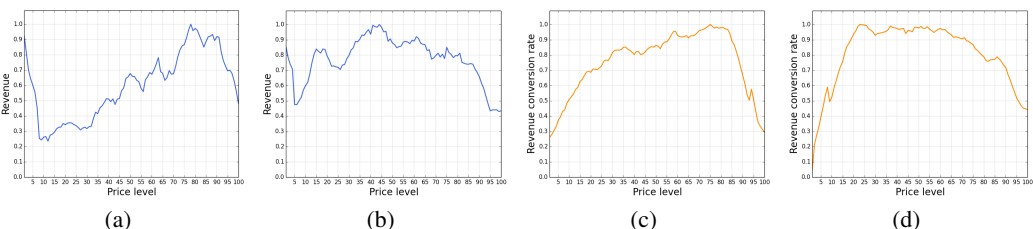

Figure 2: (a) and (b) show the average revenue for 3300 SKUs of shampoos and 4800 SKUs of candies respectively in different price levels from 1 to 100 within 3 months. Price level 1 stands for the lowest price within 3 months and price level 100 stands for the highest. (c) and (d) show the average revenue conversion rate for shampoos and candies respectively. The numerical value for revenues and revenue conversion rates are rescaled by dividing the maximum values.

As shown in Figure 2 that, revenue conversion rate is more convex than revenue itself. In the online experiments, using revenue conversion rate as reward function works fine in markdown pricing application when 1) there is a very clear and accurate stock determining the life-cycle of the pricing process; 2) most of the markdown products are low-sales-volume luxuries, having low but sensitive revenue conversion rates with prices. However, in another pricing applications in this work, daily pricing for fast moving customer goods (FMCGs), supply is adequate and the stock could be regarded as unlimited. A clear end point for each pricing process could hardly be defined. More importantly, the average sales volumes for these FMCGs are lot higher than the luxuries in markdown pricing. In this case, despite we find the relationship between prices and average revenue conversion rate through a certain period, this relationship may not keep steady at different time in this period. Another investigation for two different kinds of products about the trends of their price level with revenue conversion rate for 90 days (shown in Figure 3) reveals this phenomena.

We could see from Figure 3 (a) that, for these luxuries, when the price drops, the revenue conversion rate goes up, especially around day 25, 45, 65 and 85. However, for FMCGs in (b), there is no

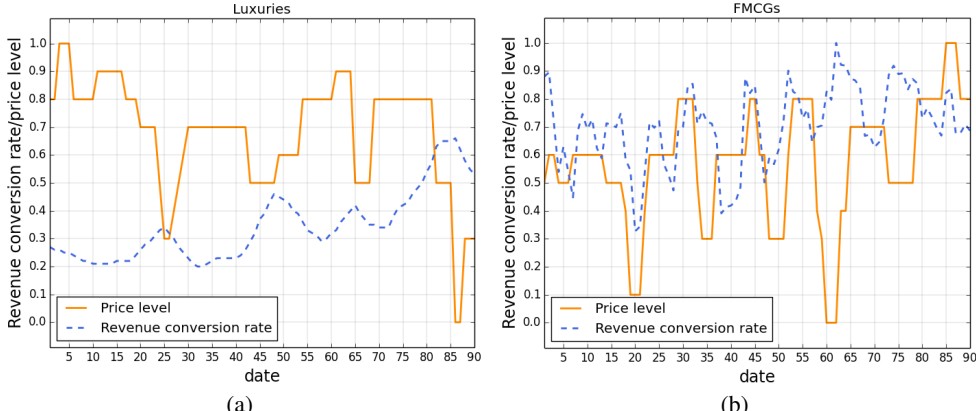

Figure 3: the average revenue conversion rate and price level for 2000 SKUs of luxuries (in sub-figure (a)) and 4000 SKUs of FMCGs (in sub-figure (b)) respectively through 90 days. 0 stands for the minimum value through 90 days and 1 stands for the maximum value, for both revenue conversion rate and price level. Correlation coefficients for luxuries and FMCGs are -0.57 and 0.15 respectively.

such relationship, but the revenue conversion rate fluctuates with an individual frequency around a week. Therefore, with the revenue conversion rate used as reward function for these FMCGs, the convergence of the model could not be guaranteed. Comparing two phenomenon in Figure 2 and Figure 3, we define a different reward function, using the difference of the revenue conversion rates (DRCR) in Eq.(1).

$$r_{i,t} = \frac{revenue_{i,t}}{uv_{i,t}} - \frac{revenue_{i,t-\tau}}{uv_{i,t-\tau}} \tag{1}$$

where $\tau$ represents the length of the time to compare the revenue conversion rate. The idea behind this definition is that, we hope to give the agent an positive signal, if it successfully raise the revenue conversion rate with its pricing action. This definition of reward function solves the convergence problem in FMCGs daily sales, while could also work out fine in markdown pricing.

## 3.2 DISCRETE PRICING ACTION MODELS

In this part, we start to introduce the models we use for solving dynamic pricing MDP we defined above. We first use Q-learning (Watkins (1989)) to find the optimal policy. Q-learning is a value iteration method to compute the optimal policy. It starts with randomly initialed $Q$ value and recursively iterates using the transitions $t = (s, a, r, s')$ to get the optimal $Q^*$ as well as the optimal policy

$$Q_{t+1}(s,a) \leftarrow (1-\alpha) \cdot Q_t(s,a) + \alpha \cdot [r + \gamma \cdot \max_{a'} Q_t(s',a')] \tag{2}$$

Here $\alpha \in (0, 1]$ is the learning rate. $\gamma$ is the discount factor. Due to the high dimension of the state space, we use a deep network to map the Q-values from the state space, which follows the idea of deep Q-networks (DQN, Mnih et al. (2015)). To update the action value network, a one-step off-policy evaluation is used to minimize the loss function:

$$L(\theta) = \mathbb{E}_{(s,a,r,s')\sim D}[r + \gamma \cdot \max_{a'} Q(s',a'|\theta') - Q(s,a|\theta)]^2 \tag{3}$$

Here $D$ is a distribution over transitions contained in a replay buffer working. $\theta$ are the parameters of the Q-network and $\theta'$ are the network parameters used to compute the target. The target network parameter $\theta'$ are only updated with the Q-network parameters every C steps. These are the two main techniques for DQN, experience replay and separate target network from Mnih et al. (2015).

To apply this discrete action method, we divide the pricing space from $P_{i,min}$ to $P_{i,max}$ into $K$ separated areas, as $K$ discrete actions. The price $p_{i,k}$ satisfies the inequalities:

$$P_{i,min} + \frac{(P_{i,max} - P_{i,min})}{K} \cdot (k-1) \leq p_{i,k} < P_{i,min} + \frac{(P_{i,max} - P_{i,min})}{K} \cdot k \tag{4}$$

will be regarded as choosing the $k$th ($k \in [1, K]$) pricing action for product $i$.

### 3.3 Continuous Pricing Action Model

Pricing on discrete action space encounters an obvious conflict setting the number of discrete actions, the hyperparameter $K$. If $K$ is too small, a large pricing area will be regarded as the same price. At the same time, algorithms with discrete action space could only output a subspace of the pricing action space, which could be unclear if the total action space divided into few subspaces. On the other hand, if $K$ is too large, a lot of actions will not be explored in the history and the exploration in the future could also be inefficient. Therefore, we consider to build up model pricing on continuous space to output an exact price instead of a pricing area. We apply the actor-critic algorithm (Witten (1977)), which combines value-iteration methods and policy-iteration methods and have been proposed and performed well on different problems(Vamvoudakis & Lewis (2010), Mnih et al. (2016)). Specifically, we apply deep deterministic policy gradient (DDPG, Lillicrap et al. (2015)) as our actor-critic method. The actor part of this model maintains a policy-network $\pi(a|s; \theta^\mu)$, taking the environment state as its input and output continuous actions $a = \mu_\theta(s)$. And the critic part takes both the state and action as input and estimates the action value function $Q(s, a|\theta^Q)$. $\theta^\mu$ and $\theta^Q$ are the network parameters. So, the loss function would be:

$$L(\theta) = \mathbb{E}_{(s,a,r,s')\sim D}[r + \gamma \cdot Q(s', \mu(s'|\theta^{\mu'})|\theta^{Q'}) - Q(s, a|\theta^Q)]^2 \tag{5}$$

Here we also apply the experience replay and separate target network techniques, so $\theta^{\mu'}$ and $\theta^{Q'}$ are the target-network parameters for actor and critic respectively. And it takes the gradient of Q-value to update the policy-network:

$$\nabla_{\theta^\mu}\mu \approx \mathbb{E}_{\mu'}[\nabla_a Q(s, a|\theta^Q)|_{a=\mu(s)}\nabla_{\theta^\mu}\mu(s|\theta^\mu)] \tag{6}$$

The idea behind is to adjust the parameters $\theta^\mu$ of the policy-network in the direction of the performance gradient, underlying these algorithms is the policy gradient theorem (Sutton et al. (2000)). In this way, the actor of our model would output a specific price on the continuous pricing action space rather than output a price area, while the critic of the model will evaluate this specific action, which could improve the effectiveness and accuracy of the DRL model.

### 3.4 Pre-training

If we directly apply reinforcement learning algorithms to E-commerce dynamic pricing, they will meet the cold start problem, starting with very poor performance and may cause capital loss. In some other areas like robotics Levine et al. (2016) and games Silver et al. (2016), there may be accurate simulators, within which the agent could learn policy. However, there is no such a simulator for dynamic pricing problem. Instead, we have enough data of the environment and the pricing decisions made by some previous controllers. These controllers could be some specialists or some rules, and some of their pricing decisions may be reasonable. The records of their decision facing the environment could be regarded as demonstrations, which has been proved to be effective for pre-training the agent from Sendonaris & Dulac-Arnold (2017).

As mentioned before, the pricing actions were taken periodically. Thus, the state of the environment as well as the rewards could be represented by the data collected within these periods between actions. Therefore, we form the demonstration in tuples $< s_t, a_t, r_t, s_{t+1} >$ and use them for pre-trianing. Specifically, we refer the ideas of Deep Q-learning from Demonstration (DQfD) Sendonaris & Dulac-Arnold (2017) as our pre-training method for DQN and Deep Deterministic Policy Gradient from Demonstration (DDPGfD) Vecerík et al. (2017) for DDPG.

### 3.5 Offline Evaluation Methodology

As we discussed above, we need to evaluate the model with demonstrations during pre-training before pricing online. W We will introduce the methodology for offline evaluation in this part, while the methods for online evaluation will be discussed in detail in section 4.2. We firt use the latest $T$ periods of records to from tuples $< s_t, a_t, r_t, s_{t+1} >, t \in [1, T]$. And then, we divide these tuples into two parts: the first D tuples will be used for pre-training, where $t \in [1, D]$. And for $D < t < T$, the tuples will be used for evaluation. The first part is to evaluate the ability to get reward for different models. The idea is that we sum the reward $r$ only if the action $a$ is close to the output of the policy $a - \epsilon < \pi(s) < a + \epsilon$. The detail of the evaluate algorithm is sketched in Algorithm 1.

---

**Algorithm 1** Policy Evaluation with Demonstration Tuples.

---

**Input:** $T > 0$: number of demonstration tuples for evaluation; $\pi$: the policy to evaluate;
**Output:** $R_\pi$: average reward form policy $\pi$;
1: $R = 0, N = 0$
2: **for** step $t \in \{1, 2, ..., T\}$ **do**
3:     **repeat**
4:         Get next tuple $< s, a, r, s' >$
5:     **until** $a - \epsilon < \pi(s) < a + \epsilon$
6:     $R \leftarrow R + r$
7:     $N \leftarrow N + 1$
8: **end for**
9: **if** $N > 0$ **then** $R_\pi = R/N$
10: **else** $R_\pi = 0$

---

Then we evaluate the accuracy of the model. For discrete action space method, DQN, we use Eq.(2) to calculate the expected immediate reward between two state $s_t$ and $s_{t+1}$ from the model, and then compare it with the real reward $r_t$ to get the error rate $e$:

$$e = |\frac{\mathbb{E}[r_t] - r_t}{r_t}| = |\frac{Q(s_t, a_t) - \gamma \cdot \max_a Q(s_{t+1}, a) - r_t}{r_t}| \quad (7)$$

While for DDPG, Eq.(7) should be changed to Eq.(8):

$$e = |\frac{Q(s_t, a_t) - \gamma \cdot Q(s_{t+1}, \mu(s_{t+1})) - r_t}{r_t}| \quad (8)$$

Notice that, we set a constant threshold $0 < r_c << 1$ for calculating errors here. If $|r_t| < r_c$ the real reward is too small, then error is defined with $e = |\mathbb{E}[r_t] - r_t|$.

## 4 EXPERIMENTAL RESULTS

We mainly evaluated the two DRL methods introduced above for dynamic pricing: DQN and DDPG. We first introduce the offline experiments, using data from Tmall.com. Then we introduce the online experiment results where we changed the online prices for products on Tmall.com in a markdown scenario and a daily basis.

### 4.1 OFFLINE EXPERIMENTS

In the offline policy evaluation, we chose 40,000 SKUs of fast moving consumer products with 60 days selling records forming about 2,400,000 tuples in total. We used the first 59 days' records as demonstrations for pre-training and the last day's for evaluation. We set $K = 10$, $\alpha = 0.01$, $\tau = 1$ and gradually increased $\gamma$ from 0.5 to 0.99. For DDPG policy evaluation, we set $\epsilon = 0.05$. The result is shown in Figure 8(a) (in Appendix). We could see that, DDPG performs better than DQN after pre-trainied with a certain number of demonstrations. Then we did the experiment for testing the accuracy of different models during the pre-training. The result of the experiment about the accuracy discussed in section 3.5 are shown in Figure 8(b) (in Appendix). It shows that, DDPG has lower error rate than DQN during pre-training.

### 4.2 ONLINE EXPERIMENTS

**Pricing for markdown season.** We first applied DQN for pricing around 500 SKUs of luxury products (mainly handbags and clothes) during the markdown season. Each product has around 10 items in stock and the aim of the markdown season is to maximize both the profit conversion rate and the revenue conversion rate. There is another group with 2000 SKUs of similar products pricing manually in the same period, which could be regarded as the benchmark to exclude the influence of the environment. Here in this experiment we set $D = 90$, which means that we used 90 days historical data for pre-training. The result of the experiment is shown in Figure 4. In the first 15 days of this online experiment, these products were in their daily prices. It shows that two groups

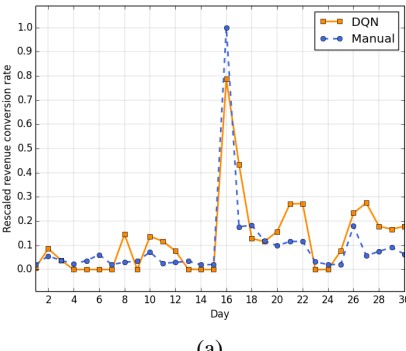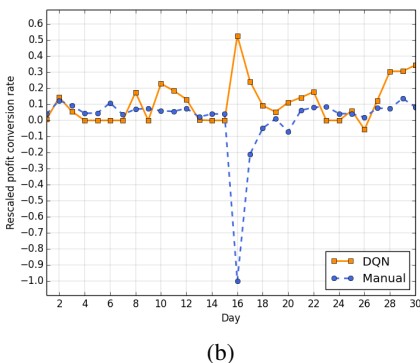

(a)                                                    (b)

Figure 4: (a) and (b) are the rescaled revenue conversion rate and rescaled profit conversion rate plots comparing products priced by DQN and products priced manually. From Day 1 to Day 15, products were in their normal sales prices. DQN group and manual group both had a revenue conversion rate of 0.04, and a profit conversion rate of 0.06. Markdown pricing started from Day 15 and ended at Day 30. DQN group achieved a revenue conversion rate of 0.22 on average and manual group's conversion rate is 0.16 for this period (with Day 16 manual group's revenue conversion rate rescaled to 1). In (b), during the markdown season, DQN group obtained a profit conversion rate of 0.16, while manual group's profit conversion rate dropped to -0.04 on average (with Day 16 manual group's profit conversion rate rescaled to -1).

of products performs alike in daily sales. Then the markdown season started at the 16th day and lasted for 15 days. We could see that both groups boosted the revenue conversion rate at day 16 and dropped sharply in the following two days. It could be caused by the so-called *pre-heat phase*: one or two days before activities, during which products are showing with the prices for activities but still selling in their daily prices. Therefore, day 16 illustrates a very common phenomena in E-commerce activities called *explosive phase*, normally at the beginning of the activities contributing the main revenue. Then in day 21 and 22 as well as day 25 to 30, DQN group successfully pull up the revenue conversion rate again beating manual group due to the policy aiming maximize the total revenue conversion rate. Comparing with the profit conversion rate graph Figure 4b, it is more clear that, manual pricing method pulled up the revenue simply by decreasing the prices lower than the cost, causing negative profit while DQN pricing policy successfully boosted the revenue while keep positive profit at most of the markdown season. It is interesting that, at day 26, DQN also priced the products to negative profit rate, and got negative immediate reward. But this action pulled up both the revenue and profit conversion rate in the rest of the markdown season.

**Pricing for daily sales.** To verify the effectiveness of reward function in Eq.(1), we set up another online experiment, to price supply unlimited FMCGs. We first defined *simi-products* for E-commerce platform, the products with the same brand, same category and similar selling behaviours. We have found that, using the same pricing policy, even if two groups of simi-products have different total revenue, their DRCR could be very close (Figure 5(a)), due to the elimination of the season fluctuation and influence from management strategy. Therefore, we could evaluate different pricing policies by comparing their DRCR. We chose around 1000 SKUs of FMCGs from Tmall.com, mainly foods, snacks and daily chemicals, as our experiment group (group two in Figure 5(a)), and then matched 3000 SKUs of simi-products also selling on the platform as the control group (group one in Figure 5(a)).

In the first 20 days of experiment, we investigated the behaviour of DQN pricing policy by comparing it with the control group. We set $D = 30$, shorter than the markdown pricing as the FMCGs' behaviour changes more rapidly than the luxury products'. Other parameters were set same as the off-line evaluation. DRCR within 20 days for two groups are shown in Figure 5(b). We could see that, DQN group outperformed control group.

Then we divided the experiment group randomly into two groups for testing DQN and DDPG and used the same parameters as off-line evaluation. The result is shown in Figure 5(c). During this part

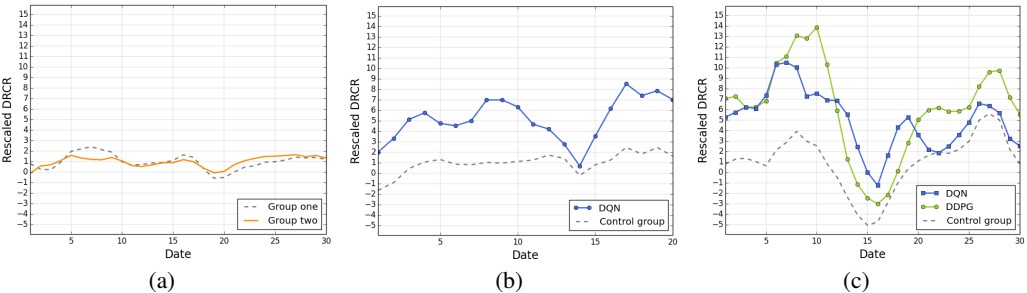

(a)            (b)            (c)

Figure 5: Comparing the DRCR for different groups of similar products. (a) shows two groups of similar products which have similar DRCR within 30 days. The average for group one is rescaled to 1.00 and the average for group two becomes 0.99 after rescaling. (b) shows the DRCR for similar product groups priced by DQN and control group respectively for 20 days. The averages of DQN groups is 5.24, with the average of the control group rescaled to 1.00. (c) shows DRCR for similar product groups priced by DDPG, DQN and control group for 30 days. The averages of DDPG and DQN pricing groups are 6.07 and 5.03 respectively with the average of the control group rescaled to 1.00.

of experiment, we encountered some daily management activities (with some coupons given). These activity only caused less than $10\%$ fluctuation of the total revenue and the phenomena of DRCR we mentioned above could still be observed. Two DRL methods both outperformed control group while DDPG performed better.

Referring Ibrahim (2013), we assessed the importance of the input state features from trained neural networks. We used the Connection Weights Algorithm (Olden & Jackson (2002)) to calculate the relative importance for all the features we introduced in section 3. The importance scores for the four groups of features introduced in section 3 are shown in Figure 6, with the highest score rescaled to 1. We could see that, the features with highest relative importance are the UV related features and some related prices, while the lowest is the present sales volume, revenue and number of buyers. In general, customer traffic features and price features get higher scores while the sales features get lower scores. Comparing with the investigation in section 3, it gives us an insight for dynamic pricing problem on E-commerce platform about the features influence to the pricing results. An interesting phenomena is that, in the group of competitiveness features, monthly average score for the products gets higher importance than weekly average score, while weekly comments gets higher importance than monthly comments. This is because the scores linked to a product showing on the page is calculated monthly, while most of the customers may only read the latest comments. The explanation for input features in Figure 6 is given in Table 1 (in Appendix).

## 5 CONCLUSIONS AND DISCUSSION

In this work, we proposed a deep reinforcement learning framework for dynamic pricing on E-commerce platform. We defined the pricing process as a Markov Decision Process and then defined the state space, discrete and continuous action space, and different reward function for different pricing applications. We applied our methods for pricing policies and applied to online pricing in real time. We first apply deep reinforcement learning methods for pricing products in a markdown season. The field experiment showed that it outperformed the manual markdown pricing strategy. As daily pricing for FMCGs, we design a systematic mechanism for online pricing policy evaluation, to address the legal issue of A/B testing for different pricing strategies. We showed that pricing policies from DDPG and DQN outperformed other pricing policies significantly.

This work is the first to use deep reinforcement learning for dynamic pricing problem on E-commerce platform, pricing thousands of SKUs of products in real-time. In this work, there are a few constraints could be removed. First, our pricing framework trains each product separately. As a result, the low-sales-volume products may not have sufficient training data. This could be solved by clustering similar products and using transfer learning to price the products in the same cluster. Meta-learning may also help for this problem. Second, our framework outputs pricing policy for

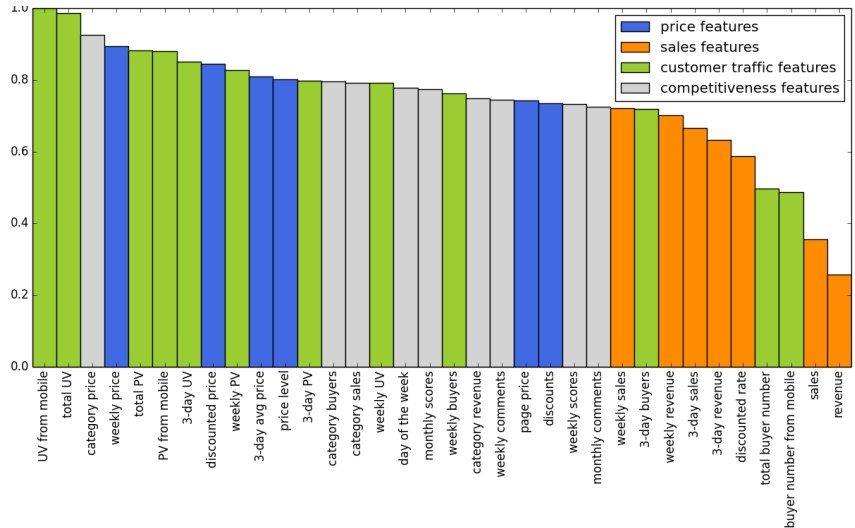

Figure 6: rescaled importance scores for some features. The average of the scores are 0.81, 0.56, 0.79 and 0.70 for price features, sales features, customer traffic features and competitiveness features respectively.

each product separately, however, sometimes we hope to price different products together to form certain marketing strategies. This may be solved by a combinatorial action space. Third, in our pricing framework, we take only the features related to the products to describe the environment state. In the future, we would try to take more kinds of features into consideration for pricing under more specific scenarios, e.g., promotion pricing or membership pricing.

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

# A ADDITIONAL EXPERIMENT RESULTS

Here we present the experiment results introduced in Section 4.1.

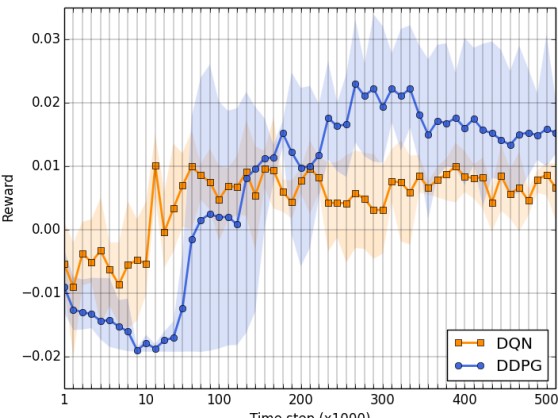

Figure 7: offline policy evaluation for DQN and DDPG with Tmall.com historical sales data.

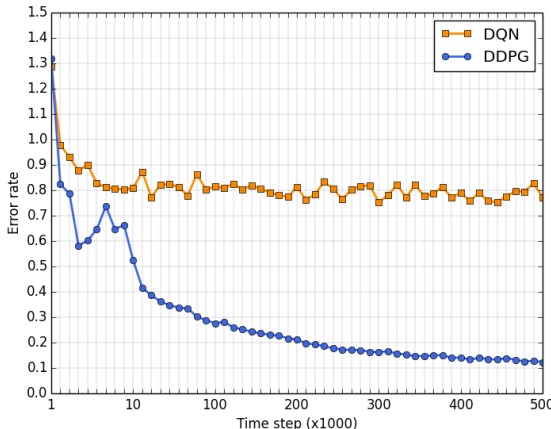

Figure 8: the error rate of the DRCR for DQN and DDPG during pre-training with Tmall.com historical sales data.

## B ADDITIONAL EXPLANATION FOR STATE FEATURES

We give some explanations for the state features introduced in section 3 here. Notice that, it only lists the features showed in Figure 6. We choose these features to just represent some mostly used data on E-commerce platform in this work and a comparison between them.

Table 1: Explanation for some state features

| feature | explanation |
| --- | --- |
| UV from mobile | unique visitors viewed the product form mobile devices |
| total UV | total number of the unique visitors viewed the product |
| category price | the average price for all the products in the category |
| week price | the average price for a product in a week |
| total PV | total time the product's detail page been viewed |
| PV from mobile | the time the product's detail page been viewed by mobile devices |
| 3-day UV | the average UV for the last three days |
| discounted price | the price actually payed by the buyer |
| weekly PV | average PV for the last week |
| 3-day price | average price for the last three days |
| price level | relative price leveled from 0 to 1 |
| 3-day PV | the average PV for the last three days |
| category buyers | the average number of buyers for the category |
| category sales | the average sales for the category |
| weekly UV | the average UV for the last week |
| monthly scores | the average scores given for the product from the buyers in the last month |
| weekly buyers | the average number of buyers for the last week |
| category revenue | the average revenue for the category |
| weekly comments | the number of comments given for the product from the buyers for the last week |
| page price | the price without discounted showing on the detail page |
| discounts | the average of discounts given to a product |
| weekly scores | the average scores given for the product from the buyers in the last week |
| monthly comments | the number of comments given for the product from the buyers for the last month |
| weekly sales | the average sales for the last week |
| 3-day buyers | the average number of buyers for the last 3 days |
| weekly revenue | the average revenue for the last week |
| 3-day sales | the average sales for the last three days |
| 3-day revenue | the average revenue for the last three days |
| buyer number | the number of the buyers for the product |
| sales | the sales-volume for the product |
| revenue | the revenue for the product |

