# OpenReview forum: "Dynamic Pricing on E-commerce Platform with Deep Reinforcement Learning"
_ICLR.cc/2019/Conference_

### Official Review · AnonReviewer2 · 2018-10-24
**Dynamic Pricing with RL**

**Rating:** 4
**Confidence:** 3

**Review:**

In this paper, the authors study the problem of Dynamic Pricing. This is a well-studied problem in Economics, Operations Research, and Computer Science. The basic problem is to find the right price for a product based on repeated interaction with the market. The significant challenge in this problem is to figure the right set of prices without knowing the future demand. The authors in this paper propose a deep reinforcement learning based approach to tackle this problem. Algorithms to this problem in the past have made assumptions such as concavity of the demand curve to make it tractable mathematically. In this paper, they do not make assumptions (although they do not provide theoretical guarantees) and one of the contributions of this paper is to handle the full generality of the problem. The other contribution is to test their methods in both offline and real-time setting and compare it against a natural MAB style algorithm (LinUCB to be precise). I have several comments on this paper.

First, the paper is not well-written. To begin with, the authors do not formally define the Dynamic Pricing problem. In literature, there are many versions of this problem. Is there limited supply or is the supply unlimited? Should we find the price for a single item or multiple items separately? It is unclear. There are pieces of information across the paper which indicate that the seller is selling multiple products for which the algorithm is pricing separately. I am still unsure about whether the supply is finite or not. Along these lines, the authors should spend some time reading through the draft carefully to fix many grammatical mistakes throughout the paper.

Second, the authors have missed out some critical related work, both in the related work section *and* in their experimental comparison. The paper Dynamic Pricing with limited supply [Babiaoff et al EC 12] and Bandits with Knapsacks (BwK) [Badanidiyuru et al FOCS 2013] provide MAB style algorithms for this problem. These algorithms overcome the reasoning they provide in their experiments -

"We noticed that, LinUCB outperformed DRL methods in offline policy evaluation. One of the main reasons is that, MAB methods aim to maximize immediate reward while RL methods maximize long-term reward".

The BwK algorithm precisely optimizes for long-term rewards under limited supply constraints. I think it will be useful to add experiments that compare the current approaches of the paper with this MAB algorithm. Overall I find the baselines in this paper to be weak and suggest more experiments and comparison against tougher relevant benchmarks. In fact, they have failed to compare, in any meaningful way, to a long line of work on Dynamic Pricing (the two Besbes and Zeevi papers they have cited for instance). I wonder how the experiments compare against those algorithms. Or they should clearly state why such a comparison is not relevant.

Overall I find the goal to be interesting and novel. But I think the current state of the draft and the experiments make it weak. In particular, the paper falls below the bar on clarity and quality in this current state.

---

> ### Author Response · Authors · 2018-11-28
> **Response to Reviewer2**
>
> Thanks for the reviewer’s detailed and thorough review. These comments are very valuable to us. We take these comments seriously and listed our response to these comments one by one:
>
> Q1: First, the paper is not well-written. To begin with, the authors do not formally define the Dynamic Pricing problem. In literature, there are many versions of this problem. Is there limited supply or is the supply unlimited?
>
>  A1: The definition of the dynamic pricing problem is now clearly proposed and added in the paper. For one pricing application, markdown pricing, the supply is limited. For another application, daily sales, the supply is unlimited. Because in this part, we are addressing the fasting moving customer goods pricing strategy, that means the inventory will be replenished when the inventory level drops below a certain value and the service level is usually fairly high, e.g. 95%, so the supply is unlimited.
>
> Q2: Should we find the price for a single item or multiple items separately? It is unclear. There are pieces of information across the paper which indicate that the seller is selling multiple products for which the algorithm is pricing separately.
>
> A2: For the time being, we are trying to find prices for single item separately and sequentially, although the prices will be effective online at the same time. To find the price for multiple items simultaneously is our next step of research.
>
> Q3: I am still unsure about whether the supply is finite or not.
>
> A3: see point 1.
>
> Q4: Along these lines, the authors should spend some time reading through the draft carefully to fix many grammatical mistakes throughout the paper.
>
> A4: The grammar issues have been fixed one by one.
>
> Q5: Second, the authors have missed out some critical related work, both in the related work section *and* in their experimental comparison. The paper Dynamic Pricing with limited supply [Babiaoff et al EC 12] and Bandits with Knapsacks (BwK) [Badanidiyuru et al FOCS 2013] provide MAB style algorithms for this problem. These algorithms overcome the reasoning they provide in their experiments - "We noticed that, LinUCB outperformed DRL methods in offline policy evaluation. One of the main reasons is that, MAB methods aim to maximize immediate reward while RL methods maximize long-term reward".
>
> A5: Thanks for providing these 2 papers that we missed mentioning in our paper. We carefully reviewed these 2 paper and found both of them address the problem with limited supply. As mentioned in point 1, the commodities are assumed to have unlimited supply, as a result, we doubt it might be unfair to compare them together. However, we carefully reviewed the reasoning provided in our paper, and found it inappropriate for this explanation. The main reason should be that RL captured the effect of state transitions in pricing strategy however MAB does not take the state transition into consideration in nature. This would be the major advantage RL can provide. What is more, the comparison of DRL with MAB is not our main goal of this paper. The main purpose of our paper is to show that DRL can outperform pricing strategies maintained by human beings, which have been clearly reflected via results provided by our paper.
>
> Q6: The BwK algorithm precisely optimizes for long-term rewards under limited supply constraints. I think it will be useful to add experiments that compare the current approaches of the paper with this MAB algorithm. Overall I find the baselines in this paper to be weak and suggest more experiments and comparison against tougher relevant benchmarks. In fact, they have failed to compare, in any meaningful way, to a long line of work on Dynamic Pricing (the two Besbes and Zeevi papers they have cited for instance). I wonder how the experiments compare against those algorithms. Or they should clearly state why such a comparison is not relevant.
>
> A6: We actually consider to try the work long the line of the data-driven optimisation methodologies to address our problem. However, we found all the papers assume the revenue rate r(lamda) concave while it is not guaranteed in E-commerce environment where the traffic volume is impacted by marketing strategy and recommendation systems. There are data provided in our paper reflecting these facts. That is why we give up these methodologies and do not conduct these comparisons.

---

> > ### Comment · AnonReviewer2 · 2018-11-29
> > **Thanks!**
> >
> > Thanks for the detailed response, and the updated version.

---

### Official Review · AnonReviewer1 · 2018-11-03
**Review for Dynamic Pricing on E-commerce Platform with Deep Reinforcement Learning**

**Rating:** 4
**Confidence:** 5

**Review:**

The authors proposed a deep reinforcement learning for dynamic pricing problem. The major contribution is on the problem formulation and application end. From the algorithm point of view, the authors adopt the existing deep reinforcement learning algorithms like DQN and policy gradient. The experiments are conducted both online and offline using dataset from Tmall.

[Advantage Summary]
1. A very interesting application to apply deep RL on dynamic pricing.

2. Experimented on industry dataset based real users. Both online and offline experiments are conducted.

[Weakness Summary]
1. This is not the first work to apply deep reinforcement learning for dynamic pricing problem as claimed by the authors.

2. Limited technical contribution.

3. Illustration and analysis of the experiment can be further approved.

[Details in weakness and questions]

1. This is not the first work to apply deep reinforcement learning for dynamic pricing problem as claimed author.
For instance,
"Reinforcement Learning for Fair Dynamic Pricing"
Since applying deep RL to dynamic pricing is one of the significant contributions of this paper, this limits the overall contribution.

2.  The technical contribution is very limited by just applying existing algorithms.

3. How to determine the step t seems to be a very important issue that can affect the performance of the algorithm and not well explained. From the experiment, the authors seem to set the period as one-day and update the price daily. But in reality, the time to update the price in a dynamic pricing system should not be a fixed value. For instance, the system should adjust the price in real time if there are changes in the environment(e.g., demand-supply change)

4. Experiment part needs more analysis. For instance, day 16 seems to be an outlier, and conversion rate drops dramatically in the following days. Why? How is the conversion rate from day 16 gets calculated in the final evaluation?

---

> ### Author Response · Authors · 2018-11-28
> **Response to Reviewer1**
>
> We appreciate the comments and they are helpful for improving our manuscript. We have made necessary changes to respond to the comments in our draft.
>
> Q1: This is not the first work to apply deep reinforcement learning for dynamic pricing problem as claimed author. For instance, "Reinforcement Learning for Fair Dynamic Pricing"
>
> A1: Thanks for providing us this missed paper. In this paper we noticed that, it indeed used Q-learning with DNN, a network with only 1 hidden layer, if we call it a ‘deep’ neural network, as its approximator. But it did not use the experience replay or target networks, the two main contributions in DQN and the main techniques in latter DRL methods. Using neural networks as the approximator in "Reinforcement Learning for Fair Dynamic Pricing" (RLFDP for short) has been discussed and applied long before the development of DRL. Furthermore, the experiments in that work were taken in simulation. As for us, we have fully defined the DRL models, both for discrete and continuous action space, defined reward functions different from the previous work and applied all needed techniques to guarantee the convergency for the real-world dynamic pricing problem. Furthermore, RLFDP allows different prices to different customers for the same commodity at the same time, which is strictly prohibited in E-commerce. The pricing strategy in our paper only permits the same price to different customers for the same commodity at the same time, which strictly complies with the regulations of E-commerce.  And the field experiments were taken on real-world E-commerce platform with thousands of SKUs of product pricing for months. So we still insist that we are the first to use DRL in dynamic pricing on E-commerce platform.
>
> Q2: The technical contribution is very limited by just applying existing algorithms.
>
> A2: The previous DRL methods are mainly applied and contributed in areas like game theory, robotics control et al. As we insisted in 1, we are the first to fully define and apply this methodology to dynamic pricing problem on E-commerce platform and it is quite different from the previous areas. First, we do not have a simulator for the real-world E-commerce market environment and it is even hard to be quantified. Second, we proved in our work that, with previously widely used reward function in dynamic pricing, the convergency of the method can hardly be guaranteed on real-world E-commerce platform. Third, we need to deal with the cold start problem, without simulation. At last, after putting it online, we still face the difficulty for not being able to A/B test due to legal reasons. Therefore, even though we are applying existing methodology, we think we made significant contributions by fully importing it to a new area.
>
> Q3: How to determine the step t seems to be a very important issue that can affect the performance of the algorithm and not well explained. From the experiment, the authors seem to set the period as one-day and update the price daily. But in reality, the time to update the price in a dynamic pricing system should not be a fixed value. For instance, the system should adjust the price in real time if there are changes in the environment(e.g., demand-supply change)
>
> A3: Step t is indeed an important hyperparameter in dynamic pricing problem, and we agree that a non-fixed time to update the price will be more strategic to generate more revenue. However, this will hurt customer shopping experience as well as break price image seriously, if prices are changed too frequently and randomly. This again limits the implementation of DRL. Fortunately, by designing a novel reward function introduced in our paper, DRL still outperforms pricing strategies proposed by human beings.
>
> Q4: Experiment part needs more analysis. For instance, day 16 seems to be an outlier, and conversion rate drops dramatically in the following days. Why? How is the conversion rate from day 16 gets calculated in the final evaluation?
>
> A4: We add more analysis about our experiment now. As for the day 16, it is a common phenomena in activities on E-commerce platform, the so-called explosive phase. Normally it is at the beginning of the activities and contributes the main revenue, for instance, the Black Friday for Christmas shopping season or the 11th November for the Double 11 Shopping Carnival. Since it released some important information of the activity, we did not take it as an outlier. Related discussion about it in detail has been added in the draft.

---

### Official Review · AnonReviewer3 · 2018-11-05
**Interesting and relevant, but still needs work**

**Rating:** 4
**Confidence:** 4

**Review:**

This paper presents a reinforcement learning approach to dynamically set the price of items on sale on an e-commerce website, based on a state description consisting of several kinds of features: price, sales, customer traffic and competitiveness. The actions (possible prices) are constrained to lie in item-specific lower and upper bounds. Against a proposed method relying a continuous action space (an implementation of the Deep Deterministic Policy Gradient of Lillicrap et al., 2015), alternatives relying on multi-armed bandits and Deep Q Networks are evaluated. Experimental results from an online deployment are presented.

The paper is at times hard to understand. In particular, it would benefit from a thorough review of English grammar and style. For instance, the alternating descriptions between the past and present tenses (e.g. the abstract and the introduction) are quite non-natural and somewhat irritating [see specific instances in the detailed comments, below].

From an algorithmic standpoint, the paper does not introduce new methods and relies on well-known reinforcement learning techniques. The proposed methodology of applying specific RL techniques such as DDPG to pricing appears novel. However, there is an abundant literature on optimal pricing and discounting in operations research, much of it based on dynamic programming techniques, and more links to this literature could be provided.

One could question the choice of the reward function chosen (eq. 1): since it is a ratio, it is extremely sensitive to variations in the denominator, and this could severely impact convergence.

The primary weakness of the paper lies in the experimental evaluation: at least the following informations are missing to truly understand the methodological impact and economic benefits of the proposed approach:

1. A complete description of the experimental setting, i.e. how many SKUs (stock keeping units) were evaluated, over which product categories, over what time horizon. Of those, how are they distributed in the fast-mover / slow-mover plane (Syntetos et al., 2005)? What special events were material during the time period? How many of those were predicted and incorporated into the state representation?
2. One or more tables of results giving expected utility gains over baseline of the proposed methods, along with confidence intervals.
3. For an ICLR submission, there should be additional investigations as to the structure of the learned representations, at least by the DQN and the DDPG models — is the state embedding learned by the networks somewhat suggestive of economically meaningful properties of the items or the sales environment?

As such, even though the paper is interesting, it is in too early a state to recommend acceptance at ICLR.

Detailed comments:

* p. 1: many past tenses that should be in the present tense, e.g. (just in the abstract): modeled ==> models, defined ==> defines, then it introduced ==> it then introduces, were designed ==> are designed. Many other cases in the rest of the paper.
* p. 1: has draw ==> has drawn
* p. 2: Forth ==> Fourth
* p. 2: overtime ==> over time
* p. 2: is assumed, to ==> is assumed to
* p. 2: described ==> describe
* p. 3: “looks deep inside learning while earning approaches” ==> sentence not clear
* p. 3: this is not clear: “since the number of page visitors may ﬂuctuate dramatically and this could lead to non-concavity“ ==> why would it lead to non-concavity?
* p. 4: The whole paragraph before eq. (1) is not clear
* p. 5: The D in eq. (5) should be explained immediately, not after eq. (6).
* p. 5: In eq. (5), $\theta’$ is not explained: how does it differ from $\theta$ ?
* p. 5: In eq. (6), how is $\theta^{Q’}$ different from $\theta^{Q}$ ?
* p. 6: in eq. (8), the denominator r_t could be a small number, leading to a noisy error; this should be discussed.
* p. 6: Below eq. (12), these sentences are not clear: “For dynamic pricing problem, we particularly concern the outcome from changing between prices. To have a well knowledge between two prices before and after pricing.”

---

> ### Author Response · Authors · 2018-11-28
> **Response to Reviewer3**
>
> We would like to thank the reviewer for the valuable feedback and suggestions to improve this draft. We address the reviewer’s concerns:
>
> Q1: The proposed methodology of applying specific RL techniques such as DDPG to pricing appears novel. However, there is an abundant literature on optimal pricing and discounting in operations research, much of it based on dynamic programming techniques, and more links to this literature could be provided.
>
> A1: We added more links in the Literature Review now, especially the previous works using reinforcement learning for dynamic pricing problem. Both dynamic programming techniques and reinforcement learning can be used to address Markov Decision Process (MDP), which is usually used to model the dynamic pricing problem. Due to the computational efforts needed to solve this MDP problem in practice, we applied DRL techniques to solve this real-world problem.
>
> Q2: One could question the choice of the reward function chosen (eq. 1): since it is a ratio, it is extremely sensitive to variations in the denominator, and this could severely impact convergence.
>
> A2: We indeed considered this problem in our experiment using a threshold 0<r_c<<1, and if the r_t in eq.1 is too small (r_t<r_c), then the error is defined with e = |E[r_t] - r_t|. Somehow we missed it in the first version of our draft. We have now added it to our draft.
>
> Q3: The primary weakness of the paper lies in the experimental evaluation: at least the following informations are missing to truly understand the methodological impact and economic benefits of the proposed approach:
> 1. A complete description of the experimental setting, i.e. how many SKUs (stock keeping units) were evaluated, over which product categories, over what time horizon. Of those, how are they distributed in the fast-mover / slow-mover plane (Syntetos et al., 2005)? What special events were material during the time period? How many of those were predicted and incorporated into the state representation?
> 2. One or more tables of results giving expected utility gains over baseline of the proposed methods, along with confidence intervals.
> 3. For an ICLR submission, there should be additional investigations as to the structure of the learned representations, at least by the DQN and the DDPG models — is the state embedding learned by the networks somewhat suggestive of economically meaningful properties of the items or the sales environment?
>
> A3: In the first version of our graft, since it is the first work using DRL for dynamic pricing problem on E-commerce platform, we tried to focus on the problem formulation. Therefore, to follow the instruction about the suggested page limit, we had to simplify the discussion about the experiment results. In the latest version of the draft, we have reorganized the structure:
> A3.1: The complete description is now added to the draft. For the markdown pricing, there are around 500 SKUs of luxury products pricing by our models and 2000 SKUs products as benchmark group (the manually pricing group). And for daily pricing, the second part of the experiment, there are over 1000 SKUs of fast moving customer goods pricing by our model, with 3000 SKUs of benchmark. Other details are given in the Experimental Results (section 4).
> A3.2: We added some investigations before the experiments, giving the reason we defined our reward functions. Also, we added more discussions about our experiment results now.
> A3.3: We now added additional investigation with the trained DQN and DDPG neural networks about the importance of the input state features referring Ibrahim (2013). In this part of investigation, we found that, the defined price features and customer traffic features have higher importance scores than the sales features. This could be some useful links between the outcome of pricing action and market environment state features.

---

### Public Comment · (anonymous) · 2018-10-22
**The Question of the Reward Function**

According to the paper, the reward is defined as the difference of the conversion rate in the current step and that of the last step, which means that the long-term rewards are indeed equal to the conversion rate of the last step. However, in reinforcement learning, the objective is to maximize the averaged revenue conversion rate of the entire episode. Thus, the reward function defined in the paper is meaningless.  Also, I think some missing explanations in the paper are: 1) why the reward does not approach to 0 in your offline experiments 2) why the conversion rate is not stable in your online experiments.

---

> ### Author Response · Authors · 2018-10-23
> **Re: The Question of the Reward Function**
>
> Thanks for your comments. Let’s clarify some of your misunderstanding:
>
> We defined two different types of reward functions in the paper (please refer to Section 3 and 4 for details): the first reward function maximizes the total expected conversion rates overtime (with discount), which corresponds to the revenue management notion of optimality in pricing. The second reward function maximizes the total expected conversion rate differences overtime (with discount). This is a different objective in general, however these two objectives are closely and positively related, and if needed we can give their precise relationship in mathematical terms.
>
> Real world commercial environment is not stationary, as a result, the real conversion rates and their differences may not converge to zero or stabilize at a point on-line or off-line. We will think about adding more explanation in the final version.

---

### Meta-Review · Area_Chair1 · 2018-12-14
**This is an interesting topic but the reviewers had substantial concerns on the clarity and significance of the contribution.**

**Confidence:** 4
**Recommendation:** Reject

**Metareview:**


This is an interesting topic but the reviewers had substantial concerns on the clarity and significance of the contribution.